# Applications of Experience Economy in Craft Beer Tourism: A Case Study in Thailand's Context

**Rangson Chirakranont *** and **Sirijit Sunanta**

Research Institute for Languages and Cultures of Asia, Mahidol University, Nakhon Pathom 73170, Thailand; sirijit.sun@mahidol.edu
* Correspondence: rangson.chi@mahidol.ac.th

**Abstract:** With Thailand as a context, this study explores the applications of Pine and Gilmore's experience economy framework in two forms of craft beer tourism, namely brewpubs and festivals. A three-pronged qualitative research design is used wherein data was collected through observations, interviews ($n = 26$), and netnography ($n = 203$). The findings revealed that both forms of craft beer tourism implement four dimensions of the experience economy in different measures. Esthetic and escapist dimensions benefit from natural, scenic settings and a man-made environment. In addition, they advance the framework by proposing the dimension of entrepreneurship, which orchestrates the entertainment and education elements that enhance tourists' experiences. The application of the experience economy framework is a useful strategic approach for craft beer tourism which can be applied in niche or special interest tourism, while also providing a significant influence on destination marketing and sustainable development.

**Keywords:** craft beer tourism; experience economy; entrepreneur experience; Thailand





## 1. Introduction

Craft beer is produced by craft breweries which are typically small and independent [1]. The particular characteristics of craft beer include having a distinct flavor as well as a unique brand name and label [2]. The craft beer movement started in the late 1970s and has been followed by the rise of microbrewers owing to the ease of supplies and knowledge of beer brewing. Brewers around the world offer a wide variety of craft beers, which reflect the transformation of beer consumption and the entire industry. The UK, for instance, devotes itself to traditional high-quality real ales which inspired the global craft beer industry [3]. Italy and Spain's beverage consumption has shifted from wine to beer since the 1970s. The Italian craft beer industry, for example, grew 97.6 percent within four years with the number of microbreweries rising from 206 in 2008 to 407 in 2012 [4]. Europe, a region that has a long history of beer production, influences the style of craft beer in the U.S. In the United States alone, the craft beer market generated $27.6 billion in sales, which accounts for over 24 percent of the total U.S. beer market, and contributed to 7% of market growth in 2018 [5].

The growth of the craft beer movement has prompted the emergence of craft beer tourism, a recent development in food and beverage tourism [6]. Major forms of craft beer tourism include beer tasting and beer-related experiences tied to visiting breweries, beer trails, beer festivals, and events [6–8]. In the current craft beer tourism scene, the United States, Canada, the United Kingdom, Australia, and South Africa are principal players who dominate the global market [9]. Hence, most of the studies on craft beer tourism have been conducted within these contexts. Extant studies tend to cover different perspectives of craft beer tourism, ranging from tourists' motivations [10], collaboration and networking between tourism businesses [11], manipulating festivals for product distribution and sociocultural space in craft beer events [12,13], and place-making and destination management [14] to stakeholder management tied to craft beer tourism [15].

Few studies have focused on looking into craft beer tourism beyond these contexts and contents. In response, this study seeks to extend the frontiers of knowledge by examining craft beer tourism in Thailand from an experience-driven approach that supports business sustainability. By doing so, the study serves two key purposes.

First, by using Thailand as the context, this study touches upon a unique scene of the craft beer tourism movement. In Thailand, tourism plays an important role in driving the craft beer movement. Nevertheless, due to several legal constraints that favor big beer players over small brewers, limited craft-beer-related activities to promote craft beer tourism are allowed. Therefore, whereas craft beer competitions, craft beer festivals, craft beer tasting, craft beer brewing lessons, and visitations to craft breweries are common activities found in other countries to promote craft beer tourism, major craft beer tourism forms in Thailand are limited to festivals and brewery visits. How then can the craft beer tourism sector, especially small businesses, grow and be sustained in a country governed by unfavorable regulatory control?

To answer the abovementioned question, the second purpose of this study highlights the potential role of tourism experiences in driving the growth and sustainability of craft beer tourism. Taking the two dominant craft beer tourism practices in the Thai craft beer movement—namely, craft beer festivals and brewery visits—this study adopts Pine and Gilmore's [16] four domains of the experience economy framework (education, esthetic, escapism, and entertainment) to identify salient factors which can benefit brewers and festival organizers, or craft beer tourism entrepreneurs, in several ways. First, the experiential domains can help inform craft beer tourism entrepreneurs of craft beer tourists' motivational drivers and can be used in designing tourism experiences that enhance business outcomes and processes, such as customer satisfaction, loyalty, and the intention to revisit [10,11], as well as co-creation and innovation [17,18]. Second, craft beer tourism entrepreneurs who strive to promote the ultimate experience to tourists are likely to not only survive but excel in the longer term [15]. Therefore, this study attempts to address the following research questions. What experience domains are utilized in the Thai craft beer tourism forms, specifically brewery visits and craft beer festivals, and how could craft beer entrepreneurs adopt those experience domains in their business strategies for sustainable growth?

Furthermore, this study contributes further to the body of knowledge on tourism experience by adding a fifth domain to the experience economy framework. Adding to Pine and Gilmore's perspectives of the experience economy, this study proposes an entrepreneur experience. It argues that craft beer tourism entrepreneurs perform two main roles. First, they are an experience themselves to the beer tourists. Second, the entrepreneurs stage and manage other experience domains using different elements to provide a memorable experience. As documented below, the entrepreneur experience is associated with increasing a tourist's overall experience.

## 2. Literature Review

### 2.1. Craft Beer Scene and Craft Beer Tourism in Thailand

Craft beer was introduced to Thailand in about 2012 and has gained increasing popularity among Thai consumers. It received approval due to its taste and texture which was relatively new to Thai beer consumers. Soon after, a number of craft beer enthusiasts began brewing beer to their preferences. The Thai homebrew beers were not only exchanged within the group of brewers but also distributed to other consumers on several limited occasions, such as casual beer events within the community.

Thai alcohol laws constitute the key factor that regulate the craft beer scene. The Thai alcohol industry, which includes the craft beer sector, is not yet liberalized—all types of production and marketing activities being controlled by the government. Thai alcoholic beverage laws prohibit home beer brewing and only allow industrial production. Furthermore, marketing activities concerning advertising, price discounts, selling times, and areas are regulated, which hinders consumption and distribution. As explained:

*"Beer brewing is legal in only two cases–with industrial-scale production of at least 10 million liters of beer per year and registered capital of 10 million baht, or with brew-pubs which produce and sell on site of at least 100,000 L of beer per year. The registered capital of 10 million baht is also applicable to small-scale producers"*. [19]

Over the years, several determined Thai craft beer brewers have been visited and fined by the police but they have not given up home brewing. To avoid legal actions, some chose to set up contract production of their beer in foreign countries such as Cambodia and South Korea before importing the beer back to Thailand. Currently, there are more than thirty brewing companies with over forty brands that use the service of contract brewers in foreign countries to obtain the alcohol stamp [20,21].

To date, there are more than 60 Thai craft beer brands in Thailand, concentrated in Bangkok and big cities such as Chiangmai and Phuket [20]. The Thai craft beer sector has shown significant growth in market value; the record of 35 million baht in 2016 multiplied more than eightfold into approximately 300 million baht in 2017 [9]. In 2018, the proportion of market share for craft beer was valued approximately at 16 million USD, expecting a twofold expansion in 2020 [22].

The Thai craft beer sector benefits from beer tourism. Even though Thai craft beer tourism does not apply all beer tourism forms, due to the government regulations and the business environment, the available forms reveal the potential to attract a number of tourists. Craft beer events and festivals are one of the important forms of beer tourism and a driving force of the Thai craft beer movement in the past few years. In Thailand, craft beer festivals are organized in different sizes and locations. Because of the lack of proper permits and the limited number of craft beer pubs and bars, many craft brewers use craft beer events and festivals as a main distribution channel, as they can sell a high volume of product in a short period. Moreover, craft beer festivals are a special occasion that brings together brewers and craft beer enthusiasts. The craft beer tourists are motivated by various elements featured in the festival such as food, facilities, and entertainment. At the festival, tourists have the opportunity to taste craft beers from multiple breweries in one stop [23]. In turn, the brewers can explain product details directly to the consumers, which leads to the promotion of sales.

Concurrently, brewery visits are a slow yet growing tourism form in Thailand. Brewery visits in several locations attract many consumers and tourists by offering various experience attributes in the premise. For instance, tourists are able to taste a variety of unique beers while they enjoy the surrounding atmosphere at the brewery [6,14]. Moreover, brewery visitations present specific knowledge, such as the beer-making process, ingredients used, and different types of beer, to the visitors [24]. Since the Thai government set the high requirement in setting up a brewery, most of the brewery visits in Thailand are in a brewpub form. The most remarkable brewpub in the Thai craft beer scene is Chit Beer (CB). CB is the first craft brewpub in Thailand, which was operated illegally in the beginning. After being visited and fined several times by the police, CB is still open and represents a part of the revolutionary craft beer movement in Thailand. Moreover, CB also runs a brewing academy that provides brewing lessons for many craft beer enthusiasts. Many students who learned beer brewing turned to craft beer entrepreneurs establishing their own brewpubs with a permit. Mitr Sam Phan, a brewpub located near Bangkok, is a collaboration of CB, one of the students, and four other brewers. Wizard Beer, another product from CB's academy, operates under a magic and wizard theme brewpub located in Pattaya. Although the number of craft brewpubs is not large and they are located only in the big cities, these brewpubs have achieved success by accommodating a number of tourists with a growing interest in craft beer consumption.

*2.2. Craft Beer Tourism as an Opportunity for Sustainable Business*

Generally, sustainability practices concern three main dimensions: environmental, social, and economic [25]. A particular understanding of sustainability tends to pivot on environmental responsibility, which connects to waste management, pollution control, and

climate change. Likewise, sustainability in craft beer production and consumption literature mostly focuses on the environmental dimension concerning energy consumption, recycling efforts, and supply chain systems [26,27]. However, in craft beer tourism research, the social and economic dimensions are equally substantial domains and have been addressed in several studies in different contexts [28,29]. For instance, extant studies showed how craft beer tourism can foster social sustainability through localized collaboration between craft breweries and local communities to create various craft beer tourism forms such as brewery visitations, beer trials, and beer festivals [30,31]. In addition, economic sustainability in craft beer tourism can emerge from fostering innovative business strategies [17].

Nevertheless, given the specific context of this study, which explores the nascent craft beer tourism scene in Thailand, the key challenge of small craft beer tourism players rests in their ability to survive and grow. From a sustainability perspective, this has led to economic sustainability being their priority concern. Similar to other small businesses, economic sustainability depends heavily on long-term business profitability and business adaptation [32,33]. To achieve the essence of profitability, a tourism business must obtain a number of customers or tourists to generate the expenses at the premises. Tourism operators should comprehend customer or tourist satisfaction and could utilize certain marketing strategies to reach such an essence. Hence, the experience economy framework applied in this study can be used to enhance the tourist's overall experience and maximize satisfaction. In addition, focusing on tourism experiences also serves as a creative approach to overcoming the legal constraints in the Thai craft beer tourism context. Since there are several limitations in Thai alcohol laws as previously mentioned, Thai craft beer tourism operators need to have the capacity to adapt by applying possible means to promote and run their businesses.

### 2.3. Experience Economy and Its Implementation in the Tourism Industry

A large number of academic scholars and practitioners have paid attention to the notion of tourism experiences for many years. The concepts of tourism experience have been generated from various perspectives. Earlier pieces of literature underline tourism experience as an exploration by engaging, accessing, and enjoying a subject that is different from everyday life [34]. Cohen [35] (p. 165) portrayed "strangeness and novelty" as important elements of the tourist's experience. Individuals perceive experience when they are exposed to an exotic environment, when they can immerse themselves in foreign surroundings, for instance. On the contrary, the authenticity of the place and activities equally provide a memorable experience to tourists. MacCannell [36] stressed that tourists seek authenticity at the destinations. In this sense, tourists strive for authentic experiences in divergent times and spaces, and until such experiences are responded to the search would move to other stages of experiences. Correspondingly, the studies from recent years use different aspects to conceptualize the tourism experience. Ureily [34] (p. 209) pinpointed that the development of the tourist experience has been shifted from "differentiation to de-differentiation", and from "generalizing to pluralizing". In this regard, experiences are constituted by various motivations and meanings; therefore, tourism activities should involve a variety of categories, such as cultural, social, and leisure ones. Larsen [37] utilized a psychological approach to position tourist experience into three stages, namely expectations, perception, and memory. He discussed that tourist experience results from individual cognitive processes which build long-term memory. Therefore, the aforementioned can be generalized as the fact that tourism experiences are the perception of accumulated memories from traveling to destinations or participating in particular events. The quest for tourism experience is descended from various scholars; conceptualizations of this notion are argued from different aspects, which arose from postmodernity and economy to psychology standpoints.

From a marketing viewpoint, Pine and Gilmore [16] proposed the experience economy framework for creating business opportunities in a wide range of industries, including hospitality and tourism. The framework described "the progression of economic value"

which was staged from commodities, goods, services, and experiences [16] (p. 98). Shifting from selling services to selling experiences, business operators need to think creatively and innovatively about how to deliver a memorable experience to customers. The term "experience economy" described a person perceiving experience when an individual participates in an event either physically or emotionally and comprehends a memorable experience [38]. The four domains of consumer experiences comprise entertainment, education, escapist, and esthetic (the four Es), which distinguish the level and form of customer involvement in the business offerings [16,39]. The experience economy model features four quadrants of experience domains with horizontal and vertical axes to determine the levels of involvement and connections of the tourists (see Figure 1). The horizontal axis delineates tourist participation—passively and actively. "Passive participation" is constructed when tourists are not influenced by the performance at the destination, and is characterized by the entertainment and aesthetic domains. In turn, the education and escapist domains manifest "active participation", in which individuals master some activities at the destination. With the vertical axis which reflects absorption–immersion, the tourist typically "absorbs" the presence of entertainment and education elements at a destination, or "immerses" themself by relaxedly engaging in the destination's environment through aesthetic or escapist experiences [16,39].

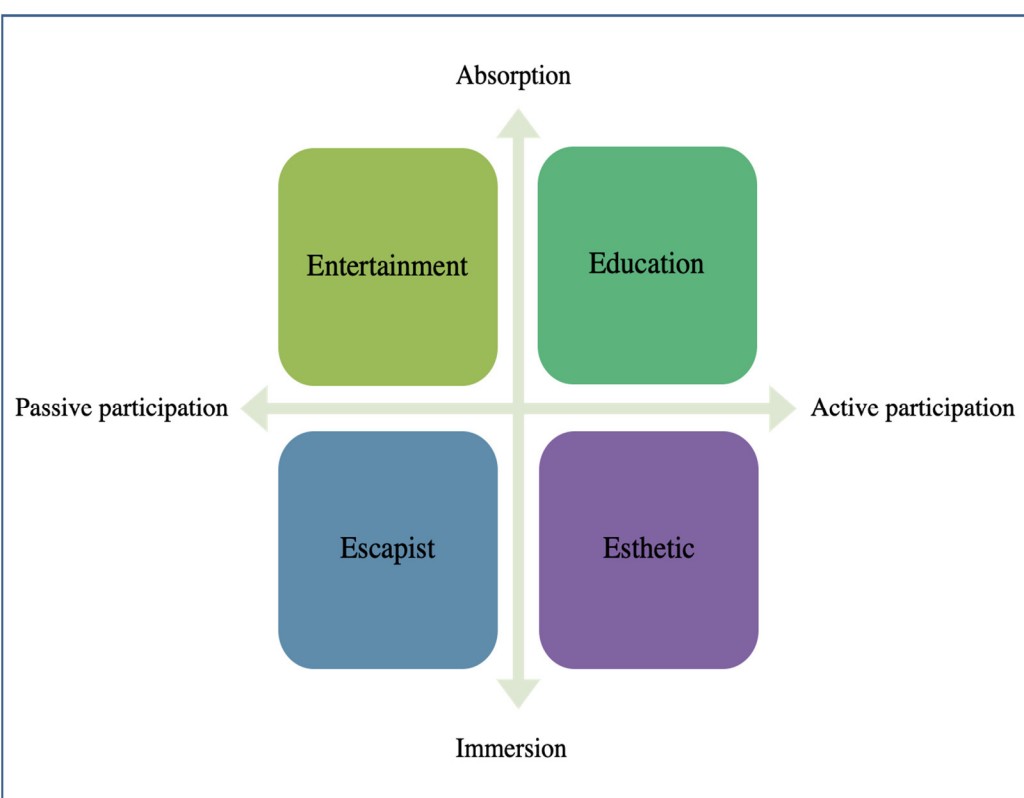

**Figure 1.** Pine and Gilmore's experience economy model.

The experience economy framework has been implemented in various hospitality and tourism studies. Oh, Fiore, and Jeoung [39] used the experience economy model to advance a measurement scale identifying tourists' experiences in lodging destinations. Shim, Oh, and Jeong [40] employed a qualitative approach to study the casino experience of the visitors. They found that visitors passively participated in gambling activities while easily immersing in a casino environment. The implication of a luxury cruise study showed that all four experiential domains played an important role in delivering positive impacts on brand prestige [41]. In food tourism studies, the framework has also been used for conceptualizing marketing and consumption in food tourism in rural

areas [42]. Wijiya, King, Nguyen, and Morrison [38] proposed a conceptual framework of international visitor dining experiences with local food, by examining the perspective of the three stages of visitors' experiences: pre-, during, and post-dining phases. Each stage involved different perceptions and expectations. In the wine tourism study, several scholars validated the model by examining tourists' experiential consumption [43] and analyzing products offered [44] in wine tourism. However, implementation has not been utilized in beer tourism. Exploring the experiential domains offered in craft beer tourism shed light on how craft beer tourism entrepreneurs exercise activities in each of the four domains, and how tourists perceive their experiences in the craft beer tourism setting. Besides, strategically designing the right experience attributes in craft beer tourism can result in higher tourist satisfaction and revenues. This study, therefore, seeks to determine the experiential domains offered in two different types of craft beer tourism, namely, craft brewpub visits in addition to craft beer festivals and events.

### 2.4. Entrepreneur as an Experience

Even though Pine and Gilmore's experience economy framework responds to four senses of tourist experience, an entrepreneur is another important factor that delivers and fortifies such experience. An entrepreneur is commonly referred to as an individual or team that utilizes resources to create business opportunities [45]. The tourism business entrepreneur is an eminent person who is directly associated with the tourist experience. Some tourists or customers emotionally connect their experience with the meeting of a famous owner [46]. It is a subjective gesture when customers are tied with "different aspects of an owner's lifestyle, interests, or role identities" [46] (p. 359). Moreover, the interpersonal interaction between owner and customers creates a positive atmosphere that provides a quality experience [47]. In this regard, the entrepreneur can be built on the experience economy framework to fulfill the tourist's total experience.

Tourists perceive entrepreneur experience mostly in small businesses rather than in big companies. This is because, in large tourism providers, customer experience is constructed by various activities performed mainly through interactions with employees or technology [48,49]. Conversely, in small business operations, many business owners tend to interact directly with customers, providing intimate and meaningful customer relations and customized service. These special interactions offer quality customer experience which associates with increasing loyalty and the strengthening of customer relationships [50], leading to overall tourist satisfaction and revisiting intention within hospitality and tourism businesses. The role of business owners in providing entrepreneur experience, however, depends on several key competencies. Gilboa et al. [48] suggested that effective communication and personal care are keys to enhancing customer experience. Effective communication benefits the business by heightening customer relationships and loyalty [51]. Through the communication, information about the product or service is provided to promote the demand for such items [52]. In addition, another driver of the customer experience concerns innovation [53]. Innovation is the novel idea and application of the business that is generated in a distinct and creative manner [54]. In fact, many entrepreneurs become more aware of the innovation practice by offering product or service differentiation to increase competitiveness in their business [55]. This study extends entrepreneur experience to the experience economy framework. It elaborates functions and elements of entrepreneur experience based on two case analyses. Furthermore, it suggests entrepreneur experience strategies that suit small businesses.

## 3. Research Methods

### 3.1. Research Design

This research draws on a qualitative study of craft brewery visitations in addition to craft beer festivals and events, which serve as the two major forms of craft beer tourism in Thailand. The qualitative research method relates to observations, interviews, and archival studies, whereas the analysis involves field notes, interview recordings, photographs, and

memos [56]. This method provides a deeper description and in-depth comprehension of different social phenomena [57]. Qualitative research is suitable for this study because it delivers a clear understanding of individuals' experiences [58]. Moreover, this method can improve knowledge of the experience domains in craft beer tourism, which leads to extending the theoretical knowledge of the experience economy in a different context [59]. The research design used a comparative case study involving one prominent beer festival and one prominent brewery. Using a comparative case study is a powerful qualitative research approach that allowed the researcher to determine the similarities and differences of two extreme cases [60] with regards to craft beer tourism experiences.

The Case Studies

Although craft beer tourism businesses in Thailand have flourished in recent years owing to the increasing number of craft beer consumers and brewers, the turnover rate of craft beer businesses is also relatively high. Many pop-up craft beer festivals were organized once or twice before being discontinued due to the lack of attractiveness to create a revisiting intention. At the same time, despite the growing number of craft brewery pubs emerging from the relaxation of state regulation, the majority are only in the early stage of establishment with uncertain chances of long-term survival.

The two cases chosen for this study were, however, selected based on two criteria. First, they serve as well-known entities—one being a prominent craft beer festival and another a prominent brewery in Thailand among members of the craft beer community and general public. Second, their proven success can be seen through business continuity—one being a craft beer festival consecutively organized for three years (but unfortunately paused by the ongoing COVID-19 pandemic), and the other representing one of Thailand's first breweries, established since 2012.

*The Thailand Beer Festival*

The Thailand Beer Festival (TBF) is Bangkok city's biggest annual craft beer event which has been organized for three consecutive years during 2017–2019. The event was initially named Bangkok Beer Fest before it changed to Thailand Beer Festival in 2018. In 2019, the three-day event was organized from November 1 to 3 at Show DC Arena, the largest outdoor entertainment in Bangkok. This evening event which takes place during 6.00 p.m. to 12.00 a.m. comprises more than three hundred craft beer brands from many countries such as the U.S., Japan, Sweden, Australia, Korea, and Canada. It essentially included various top Thai homebrew labels. The event offered several entertainment activities, such as live performances and games, as well as live music performed by famous Thai artists.

*Chit Beer*

Chit Beer (CB) is one of Thailand's first and foremost craft brewery pubs. Established in 2012, it is located in Ko Kret island, 30 km from downtown Bangkok. CB is owned and run by Wichit Saiklao, also known as Chit, the pioneer of Thai craft brewing. In October 2014, Chit opened the Brewing Academy, allocating part of the brewpub premises to provide brewing lessons to those who are interested in craft brewing. The Brewing Academy is doubled as a brewery pub, "Chit Beer", which welcomes customers who want to try Wichit's homebrew beer. The sixty-seat premise on the Chao Praya Riverside is always packed with craft beer enthusiasts. The atmosphere at CB is filled with the scent of freshly brewed craft beer, the sight of homebrew equipment, and the exchanges of knowledge and information about craft beer. Visitors enjoyed freshly made craft beer, learning about home brewery, and meeting other craft beer lovers.

*3.2. Data Collection and Analysis*

A multi-method data collection, consisting of participant observation, interviews and netnography, was employed in this study to ensure data triangulation and an all-rounded perspective of the phenomenon. First, participant observation was utilized in this study to explore the experiential domains offered by the craft beer tourism entrepreneurs and the

behaviors of the participants or tourists. Participant observation is appropriate for case study research in different facets [61]. It is a potential method for the researcher to engage in the events and activities without interfering with the participants. In addition, the researcher can identify natural or new behaviors that could occur in a particular situation. Additionally, the participant observation offers more understanding of a "participant's subculture" in specific settings, which provides greater knowledge in specific social behavior [61] (p. 59). The researcher and research assistant attended two days at the TBF which took place on 1–2 November 2019. The approximate time in the event was 3.5 h while pictures were taken and observation notes were recorded. In respect to CB, four visits were paid between October 2019 and February 2020. Similar observation manners as with the TBF were applied to capture tourists' behaviors and environment on site.

Second, to supplement the participant observation data, a total of 26 interviews and informal conversations were conducted to enhance the understanding of what was perceived from the observation, and to facilitate responses to the research questions. A purposeful sampling [62,63] was used to select interview participants in this study. This sampling strategy concerns selecting participants who have knowledge and experience on a particular phenomenon [64]. Data saturation criteria were used to determine the number of interviews with key craft beer stakeholders, namely festival organizers, craft brewpub owners, brewers, beer bloggers, and consumers. Informal conversations with the tourists were held to support the data on how they perceived the experience from the tourism settings (Table 1).

**Table 1.** Profile of interview participants.

| No. | Gender | Age | Role | Nationality |
|-----|--------|-----|------|-------------|
| I1 | Male | 48 | Craft beer brewer and bar owner | Thai |
| I2 | Male | 37 | Craft beer brewer and bar owner | Thai |
| I3 | Male | 39 | Craft beer brewer and bar owner | Thai |
| I4 | Male | 29 | Craft beer brewer and bar owner | Thai |
| I5 | Male | 48 | Craft beer brewer and bar owner | Thai |
| I6 | Male | 32 | Craft beer brewer and bar owner | Thai |
| I7 | Male | 41 | Craft beer brewer and bar owner | Thai |
| I8 | Male | 43 | Craft beer brewer | Thai |
| I9 | Male | 39 | Craft beer brewer | Thai |
| I10 | Male | 34 | Craft beer brewer | Thai |
| I11 | Male | 40 | Craft beer brewer | Thai |
| I12 | Female | 27 | Craft beer bar owner | Thai |
| I13 | Male | 28 | Craft beer bar owner | Thai |
| I14 | Female | 31 | Craft beer bar owner | Thai |
| I15 | Male | 30 | Craft beer bar owner | Thai |
| I16 | Male | 39 | Craft beer bar owner | Thai |
| I17 | Female | 35 | Craft beer bar owner | Thai |
| I18 | Male | 22 | Craft beer consumer/tourist | Thai |
| I19 | Female | 35 | Craft beer consumer/tourist | Thai |
| I20 | Male | 37 | Craft beer consumer/tourist | Thai |
| I21 | Male | 28 | Craft beer consumer/tourist | Thai |
| I22 | Male | 22 | Craft beer consumer/tourist | Thai |
| I23 | Female | 42 | Craft beer consumer/tourist | Thai |

**Table 1.** *Cont.*

| No. | Gender | Age | Role | Nationality |
|-----|--------|-----|------|-------------|
| I24 | Male | 53 | Event organizer | Thai |
| I25 | Female | 35 | Craft beer blogger | Thai |
| I26 | Female | 37 | Craft beer distributor, bar owner, blogger, and event organizer | Thai |

Lastly, the netnographic approach, via user-generated content, was used to gather online data through customers' reviews and opinions on the online forums. Netnography is a qualitative method that is popularly used in studying cultures and communities from online communications [65]. Netnography stems its root from social science ethnography, yet focuses on collecting and analyzing online information through communication activities among users [66]. Because it is similar to the ethnographic approach on the degree of immersion in the study site, but less time-consuming, this method is widely used in various tourism studies [67]. The netnographic approach does not only provide a large amount of available information but also allows researchers several advantages. For instance, the researcher can harvest meaningful information from the interactions in the online communities where individuals describe their implicit and explicit expressions [67]. Moreover, the researcher may collect data without the consent of the individual since the information is publicly posted on the websites [65]. In the case of CB, data was collected through Google reviews (GR) and Tripadvisor (TA), which were chosen for their valuable insights on tourists' perceptions in the English language form. As of October 2020, a total of 203 reviews were collected which derived from 28 reviews on Tripadvisor and 175 reviews on Google reviews. As for the TBF, there were no reviews on either Google reviews or Tripadvisor because of its characteristic feature; that it is annually organized only for three days. For this reason, data was gathered from the TBF's Facebook page (FB) and Instagram (IG) using #Thailand Beer Festival. A total of 65 reviews were acquired.

A multiple-step process (i.e., data managing; reading and memoing; describing, classifying, and interpreting; and representing as well as visualizing) suggested by Creswell [68] was used to conduct thematic analysis. Observation notes and interview records were written in the Thai language and transcribed verbatim in the following day. Online reviews were in Thai and English, and were transcribed immediately by the researcher as well. A deductive approach based on preconceived themes associated with the four realms of experiences (i.e., entertainment, education, esthetic, and escapism) was applied in coding the data from the observation notes, interviews, and online reviews. This was supplement by an inductive approach where data-driven codes were used to draw out an emergent theme (i.e., entrepreneurship). Interpretation based on rigorous readings of the findings was made by expressing important ideas to explain the phenomenon. Table 2 shows key experience themes and a coding spectrum.

**Table 2.** The key experience themes and a coding spectrum.

| Themes | Sub-Themes | Participatory Observations | | Interviews | | Customer Reviews | |
|--------|-----------|------|------|------|------|------|------|
| | | TBF | CB | TBF | CB | TBF | CB |
| **Entertainment:** *Experiences that are enhanced by absorbing or engaging in performances and activities* | | | | | | | |
| Socializing | Hanging out with family | x | x | | x | | |
| | Hanging out with friends | x | x | x | x | x | x |
| | Meeting people with the same interest | x | x | x | x | | x |
| | Meeting new friends from around the world | | x | | x | | x |

**Table 2.** *Cont.*

| Themes | Sub-Themes | Participatory Observations | | Interviews | | Customer Reviews | |
|---|---|---|---|---|---|---|---|
| | | TBF | CB | TBF | CB | TBF | CB |
| Music and games | Live music | x | x | x | x | x | x |
| | Variety of bands | x | | x | | x | x |
| | Games and prizes | x | x | x | | | |
| Other activities | Cruise on the river | | | x | | x | |
| | Contest | x | | x | | | |
| | Riding bicycle | | | x | | x | | x |
| **Education:** *Experiences that are enhanced by learning to gain knowledge or increase skills* | | | | | | | |
| Beer introduction by owner | Knowledgeable of the owner | x | x | x | x | | x |
| Beer tasting | Beer samples | x | x | x | x | x | x |
| | Tasting different beers and flavors | x | x | x | x | x | x |
| Beer academy | Brewing lesson | | | x | | x | | x |
| | Meeting with brewmasters | x | x | | x | | x |
| Beer knowledge | Understanding product details | | | x | | x | | x |
| **Esthetic:** *Experiences that are influenced by positive physical environments* | | | | | | | |
| Scenery and atmosphere | River view | | | x | | x | | x |
| | Relaxed atmosphere | x | x | x | x | | x |
| | Man-made construction | x | | x | | | |
| Space | Packed space | | | x | | x | | |
| **Escapism:** *Experiences that derive from immersing in the time, place, and product which differs from daily life* | | | | | | | |
| Weekend getaway | Spending of weekend time | | | x | | x | | x |
| Urban getaway | Getaway from the city life | | | x | | x | | x |
| | Suburban vibe | | | x | | x | | x |
| Chill out | Relaxation from the setting | x | | x | | | |
| | Immersing in the festival atmosphere | x | x | x | | | x |
| Beer offering | List of beers | x | x | x | x | | x |
| | Various choice of beer selection | x | x | x | | x | x |
| | Novelty of beers | x | x | x | | | x |
| | Great beers | x | x | | x | x | x |
| **Entrepreneurship:** *Experiences that are delivered by the owner who has direct contact with the customers or tourists* | | | | | | | |
| Owner | Being served by the owner | x | x | x | x | | x |
| | Friendliness and enthusiasm of the owner | x | x | x | x | x | x |
| | Knowledgeable and informative of owner | x | x | x | x | | x |
| | Meeting the persons who run the Thai craft beer movement | x | x | | x | x | x |

Remark: "x" indicates the application of the particular experience.

## 4. Findings and Discussion

### 4.1. Entertainment Experience

Entertainment has been a vital form of experience and used in various tourism businesses. The entertainment experience happens when tourists passively adhere to activities or performances offered at the destination. Effective entertainment must grasp and occupy the individual's attention, and allow them to participate [39]. Forms of entertainment commonly provided at the destination involve art, music, and amusement activities. Many beer festivals increase their number of visitors by offering entertainment features. Music is a typical entertainment experience provided in beer festivals, while competitions such as judging and drinking beers are also popular for beer events depending on the organizer and destination [69].

Entertainment has always been a crucial experience at festivals. Various entertainment elements were featured at the TBF, with the two main ones being music and games, in addition to socialization. First, live music was the main entertainment activity presented on the stage throughout the event. The bands and singers were selected according to the festival-goers' ages, which ranged from teenagers to young adults, and were key attractors for joining the event as well. Furthermore, the stage was used to organize a beer ambassador contest. This activity allowed each beer brand and stall to send a female beer promoter to attend the contest. Sixteen contestants participated in this activity which drew a lot of attention, cheering, and yelling from the audience. Several games were also spotted both on the stage and at beer stalls. On the stage, the activity of the game was about craft beer knowledge questions, where participants would receive a reward or souvenir if they replied with the right answer. MCs of the event were influential motivators who diligently persuaded the audience's engagement. Accordingly, similar games were brought to entertain at beer stalls with different degrees of questions and rewards. Such beer knowledge questions created a combination of education and entertainment which has been described as "edutainment" [44]. Moreover, in the game zone, various game and entertainment booths such as beer pong, shooting games, and henna tattoos were observed to provide an amusement experience at the event. These activities created a lively atmosphere and attracted many people to join. Following excerpts from social media elaborated several forms of entertainment provided in the festival.

*Let's go on the 2nd of November, they've got Earth Patravee [the singer] and beer! What a perfect combo!* (FB3)

*The rain didn't make it any less fun. Noi [the singer] was utterly powerful, every song was good and very entertaining.* (IG11)

Secondly, a festival is an event for socialization where people with the same interest come to fraternize. In particular, as craft beer festivals accumulate different stakeholders from the craft beer sector, it is also the place to meet up with craft beer community members. As one participant who owns a craft beer bar and frequently attends beer festivals mentioned:

*I have a good time when I go to the beer events. I go there to see the community, my customers, brewers, and suppliers, and to get more relationships with them. I used to only hang out in the event but now I go with the name of the bar. It's another way to tighten a relationship.* (I12)

Entertainment experiences were also emphasized at CB. However, the experience elements differed from those at the TBF. Although live music was also present, it was, however, different. Unlike festivals wherein the bands and musicians constituted a key highlight of the event, the single acoustic band playing at CB served more as background music to enhance the ambiance of the place. Due to limited space at the premise, the brewpub was mainly occupied with tables and chairs. Hence, only a small corner in the dining area was orchestrated for the band. Nevertheless, to make it fun, occasionally visitors would be invited to join the singing in return for a free pint of beer. As noted by a few participants:

*We arrived at 16:00 when the band starts playing.* (TA7)

*They have live music in the evenings . . . . not too loud and surprisingly good talent.* (TA8)

*Lovely atmosphere with live music and a mix of local and farang [foreigners].* (GR83)

An important entertainment feature at CB, however, is related to socialization. With a remarkable location, the riverside bar provides a scenic view that sets out for a wide range of visitors. Many came with friends or individuals while some came as a family with youngsters. Furthermore, due to the limited number of seats, sometimes visitors had to join a table with others, with who they made friends eventually. Thus, CB is a unique place where customers with the same interests get together and meet new friends. As comments in the social media mentioned:

*You will find a bunch of beer lovers from around the world (sometimes foreigners take over the place!) so you can expect a friendly and casual atmosphere there.* (GR120)

Overall, the entertainment dimension played an important role at the TBF, wherein various entertainment forms—light and sound, partying with friends—contributed to a positive dynamic atmosphere. This resonated with the study by Axelsen and Swan [70], which found a strong relationship between the atmosphere of a festival and increased tourists' satisfaction. Conversely, alternate and distinct entertainment elements were provided at CB. As highlighted above, a genuine ambiance tied to socio-environmental variables has made CB a special destination for fostering authentic social connections and interactions—whether in the form of fraternizing with close friends and family, meeting new and interesting strangers, or engaging with members of the craft beer community–amidst a local setting overlooking a river view and cultural landscape [71]. The limited but quality selection of craft beers offered at CB also provides for an exciting visit. Brewpubs in urban areas, which do not possess similar scenic views or socio-environmental conditions, can try to create enjoyment by introducing entertainment activities, such as beer drinking competitions, whilst instilling a friendly and lively atmosphere that enhances social ties. This is because enjoyment is an essential factor that indicates how beer tourists choose their destination, and determines the success of the business [10].

### 4.2. Educational Experiences

Educational experiences enable tourists to increase general and/or specific skills and knowledge while actively participating in events or activities at a tourism destination [39]. Several educational activities, including beer tasting, beer and food pairings, and hops or barley selection for the beers were reported in extant studies as key forms of craft beer tourism experiences [7,72]. While beer tasting and selecting ingredients for the beers can be organized in craft breweries, beer and food pairings can be done in brewpubs or restaurants that offer a beer-themed lunch.

At the TBF, two educational activities were identified: beer tasting and provision of craft beer knowledge. Generally, beer tasting is a common form of beer tourism activity found in beer events and during brewery visits [7]. A walk into the TBF exposes visitors immediately to an array of beer stalls lined up with people eager and excited to taste the beer offerings. Beer tasting was offered in every beer stall, some providing up to ten different beer styles. Consequently, the beer tasting attracted a number of visitors to crowd around the stall, which increases the chance to promote and sell the product. Substantially, craft beer knowledge was provided along with the beer tasting. Since craft beer is a relatively new kind of alcoholic beverage and is produced with different types of ingredients and flavors, delivering education on craft beer knowledge is extremely necessary. Since there are numerous beer styles introduced to the market, bestowing product knowledge encourages purchasing demand from visitors.

Nevertheless, knowledge does not derive solely from the beer stalls, but from meeting with others in the event. For instance, one participant mentioned that he received an educational experience from exchanging ideas with other festival attendants. He further explained:

*I went to the beer festival with a big group of friends. When we got there everyone separated and went to different booths. After we tried several beers, we got back together and started talking about what we just had, some were good and some were not quite . . . not only the taste but we also evaluated the color and ingredients of the beers.* (I18)

On the other hand, three prominent forms of education experiences could be identified at CB: beer introduction by the owner, beer tasting, and brewing lessons. First, a regular sight when visiting CB is the scene of Chit, the owner of CB, either standing by the bar and recommending different types of beers or walking around the brewpub introducing beers to the visitors. Unlike other brewers, Chit possesses the strong characteristics of being friendly, approachable, charismatic, and inspiring. These traits, combined with his fame as the legendary godfather of craft beer in Thailand, make the encounter between him and visitors even more meaningful. As noted by a few participants:

*Chit can speak very good English and is happy to talk with anyone about his beers and brewing beer in general.* (TA15)

*With a friendly bartender who is also the owner, you get to learn a lot about Thai craft beer.* (GR72)

*Chit Beer is a must-go, it is a legendary place. I don't only go there for the beer, but I also want to see Chit who is the icon of the Thai craft beer movement.* (I25)

Furthermore, beer tasting is also an essential experience at CB, which is similar to the beer stalls at the TBF. However, at the bar, it seemed to be busier and crowded with groups of customers waiting to order their beverages. As a consequence, the degree of education experience provided at the bar tends to focus on the product detail (e.g., ingredients used and beer flavor) rather than the beer knowledge (e.g., the brewing technique). Regarding one tourist's review, it was mentioned that:

*They change beer kegs with different flavours every few hours, but they are all worth trying out and it is guaranteed that they will have a beer that you like.* (TA7)

Additionally, CB is a prominent place for craft beer brewing lessons, as it is considered the first craft beer brewing academy in Thailand. At CB, the area is divided into a bar and kitchen, dining area, and brewing academy. The brewing academy is designed exclusively for building an educational experience. It aims to provide education for both serious and leisure homebrew by using modified kitchen equipment for the lesson instruction.

*The place also has a "beer academy" where the truly committed can learn the ins and outs of craft beer brewing.* (TA8)

*I searched for information on google and found that Chit Beer offers brewing lessons. I took the class and realized that it's not difficult like I thought. Chit makes it easy. All you need is a big pot of water adding malt and hop, filter it add yeast, and ferment you get a beer.* (I4)

In comparison, the cases of the TBF and CB show that craft beer operators in the two tourism forms—festival and brewery—have the enthusiasm to deliver educational experiences to their customers. The education experience is a salient element that is fundamentally provided through beer tasting, craft beer knowledge, and brewing lessons. Basic information on craft beer products is also an essential component. Since craft beers have a wide range of characteristics and flavors and are produced from a variety of ingredients, disseminating more understanding of product knowledge could be a positive influence on craft beer purchasing. In addition, small businesses such as craft beer pubs and bars can apply this strategy in their premises to increase the opportunity of selling their products [73]. Given that educational experience is provided by a person, it is beneficial to build a relationship between businesses and customers, or even better, between the brewers and visitors. Therefore, creating strong and long-term relationships with the customer can help build a fence to prevent potential competitors and sustain long-term profitability as well as customer retention [74].

Overall, the educational experience is presented in the absorption axis which can be interpreted as "occupying a person's attention by bringing the experience into mind" [75] (p. 31). However, delivering an educational experience to tourists requires various strategies because the level of individual experience absorption may be different. In craft beer tourism, experience absorption is a complex phenomenon and is defined by individual motivations and backgrounds [10]. To heighten the tourist's experience absorption, craft beer operators could adopt theoretical strategies proposed by Ellis, Freeman, Jamal, and Jiang [76]. They suggested that heightened "levels of relaxation and pleasure" and elimination of "active thinking" are associated with the increase of "attention, motivation, and emotion", which could enhance an individual's level of experience absorption [76] (p. 102). In this sense, easy-listening music tends to be more suitable for both tourism forms.

### 4.3. Esthetic Experience

With regards to the esthetic experience, tourists immerse themselves in a scenic environment and ambiance at the destination. Sightseeing tourist activities are commonly associated with esthetic experiences [39]. An ale trail that involves visiting several breweries in one trip, for example, provides a unique craft brewery experience wherein tourists get to tour the facilities and taste different types of beer. Tourists are also encouraged to taste the local cuisine and immerse themselves in the rural landscape scene, adding to a memorable experience [8].

An esthetic experience was identified in both the TBF and CB in different measures. Unlike the beer trails which benefited from the natural environment, the TBF offers an esthetic experience by using man-made elements. In particular, all of the construction, such as beer stalls, food stalls, and the entertainment stage, were designed and built according to the event's theme. Tak, a festival manager, explained the process:

> *We hired a team to take care of the creative design. Since we plan to organize the event every year, the stage, decorations, and decorative materials were designed to be reused.* (I24)

The TBF was designed under an industrial theme, using old shipping containers for the decoration to create a rustic feel that resonated well with craft beer. Because the event was organized at a fully functional entertainment complex, parking spaces, restrooms, public transportation, and other services were already made available. The ambiance at the event was covered with a lively atmosphere which was the result of the sound from live music, the scent of food from various food stalls, and several promotional activities from beer stalls. Correspondingly, one festival-goer commented on how she perceived the esthetic experience from this event as casual and laid back.

> *What I like the most were the friendly atmosphere and the lawn in the middle which you can lay down and relax.* (IG8)

The atmosphere and environment dimensions represented at the festival can be described as a festivalscape, which directly affects tourists' experiences [77–79]. The festivalscape refers to heterogeneous and intangible elements that stimulate an individual's sensorial receptors [80]. For instance, ambiance, sound from live music, the scent of food, and venue decoration are elements that provide an esthetic experience to attendees. Hence, the perception of experience occurs when an individual responds to those surrounding elements.

On the other hand, the esthetic dimension of CB gains its advantage from the natural environment and surrounding scenic atmosphere. CB is located at Ko Kret, which is a small island in the Chao Phraya River, north of Bangkok. The island is renowned for being a community-based tourism destination. Highlights on the island include pottery production, local Thai cuisine, and Buddhist temples. Thus, a trip to CB requires tourists to take a boat and walk through the local community. The façade of CB is the Chao Phraya River, which all customers can enjoy along with their fresh craft beer. This riverscape is one of the reasons that customers extended time at and immersed themselves with the esthetic atmosphere at CB. As several participants noted:

> *The bar is located next to the Chaopaya river which makes the atmosphere cool and easy-going.* (GR72)

> *"Once we arrived there, we were so surprised! the bar is straight on the river and it was all full of people, which was incredible because the island was looking desert!"* (TA20)

From the researcher's observations, it is astonishing to see CB being packed on every single visit. Several online reviewers, however, shared that it made the place lively and sociable. Inevitably, sometimes an overcrowded customer can impede individual enjoyment. Chit, therefore, decided to expand the pub in his own creative way that enhanced the esthetic experience. He mentioned:

> *We serve a high number of guests but sometimes the seats at the bar are not enough. Since we are by the river, I am expanding the number of seats onto a boat. It will be docked next to the existing bar and can contain around 60–70 guests.* (I1)

Beer tourists equally appreciate the esthetic experience provided at the TBF as well as CB. In wine tourism, Bruwer and Alant [81] indicated that wine tourists perceive esthetic pleasure through the winescape. With the winescape, it benefits from the locale's natural landscape that incorporates important wine-related factors, namely rural landscapes, countryside vineyards, and wine tasting rooms [82]. Dissimilarly, in Thai craft beer tourism, it is arguable that the beer tourists increase esthetic experience from both man-made and natural atmospheres. For instance, the festivalscape at the TBF assembled various man-made elements, such as music, food, and activities, to create a favorable environment that promotes a memorable experience for tourists. In addition, the riverscape at CB, which is not a beer-related factor, can also reinforce tourists' satisfaction. The craft beer business operators can apply these considerations in their business. For example, craft beer bars in urban areas that do not possess any natural scenery can design a man-made vigorous atmosphere to strengthen the customers' esthetic experience. The esthetic experience relates to all aspects of experience, therefore, increasing the esthetic dimension affects overall tourists' experience [39,44]. Consequently, the inclusion of man-made and natural elements underpins the Thai craft beerscape.

### 4.4. Escapist Experience

The escapist experience is similar to other domains of experience that provide different measures of amusement and relaxation. However, the escapist experience requires high immersion and participation where tourists diverge themselves into the difference. In addition, escapist experience in tourism is a vehicle for a tourist to escape from their daily life by providing "actual performances or occurrences in the real or virtual environment" [39] (p. 121).

According to the aforementioned experience characteristics at the TBF and CB, the escapist experience can be seen through the activities and atmosphere delivered at both beer tourism forms. At the TBF, beer tourists experienced a wide range of craft beers from numerous beer stalls, which was a conglomerate of different craft beer brands. Tasting various choices of craft beer at a particular destination did not occur in the ordinary craft beer outlets; thus, it can be a motive for tourists to attend the festival. Moreover, participating in several activities organized in the festival allow the tourist to be immersed in the festival environment and be involved in a different stage of activities. Tak denoted that the beer list and variety of beers in the festival are influential factors that can motivate tourists to attend the event.

> *The beer list is the most important factor that attracts numbers of the attendant. We must provide what they are looking for. In the beer festival, if the attendants see the beer list on the first day and are not happy, they would not come back for the second day. It indicates the success of the festival.* (I24)

In addition, a post from one festival attendant mentioned the beers, especially of those she has not tasted. Such a variety of beers allows tourists to experience new tastes which differ from their routine life.

*The beers were cheaper than any other place and I found many craft beers that I never tried before in this event.* (IG15)

Ascribed to its location, CB, on the other hand, stimulates the tourists' visitation by the environment of a suburban setting. The authentic scene in a different lifestyle and society encourages tourists' escapist experiences in which they would break away from their routine life. Furthermore, the scenic figure of the bar and surroundings create an impulse for beer tourists to take a trip to it. Concerning the operating day which opens only Saturday and Sunday, CB is a relaxing place for visitors to having a weekend getaway.

*Recommended for a lazy afternoon weekend drinking super cold beers and listening to live music.* (TA3)

*Chill out with friends with the best music and cold beer on Saturday afternoon.* (GR10)

Plus, participating in several activities such as taking a boat to the island, experiencing local cuisine, and walking through the local community initiate the tourists' involvement at the destination. Such activities attracted the city dwellers who seek an urban getaway.

*Beautiful location, out of the way but worth the trip, and probably location is a big part of how they get away with doing what they do. Viva Chit!!!* (GR8)

*The brewery is located on the island Ko Kret so it takes a little bit of work to get there. But this is one of the best getaways you can do from Bangkok's crazy life.* (TA18)

The escapist experience has been a positive influence on tourists' motivations hence it was practiced at both the TBF and CB. The TBF applied a "virtual environment" to create a pull factor in its destination [39] (p. 121), while CB implemented an authentic environment as a motivation factor for tourists who want to immerse themselves in suburban surroundings [43]. Given that escapist experiences are constructed by the following criteria: escaping from daily routine, "immersing into destination", and participating in activities, they can be applied in any craft beer operation [39] (p. 122). In this respect, craft beer businesses can provide an escapist experience in their premises regardless of whether the location is in an urban or rural area. To illustrate, a brewpub could be established in a historic building or site to recreate highly participatory and immersive involvement with the architectural structure [83].

The four domains of experience have been presented in two craft beer tourism forms in different measures. Fundamentally, the immersion and absorption in the hedonistic atmosphere and participation in particular activities have enhanced tourists' satisfaction. Based on the research findings, the craft beerscape in Thailand does not benefit only from the natural essence of suburban or rural settings but also gains pleasure by utilizing a man-made environment. Accordingly, craft beer businesses may adopt the resources they possess to generate the four Es in order to maximize the customer's experience. However, there was one significant tourist experience element that emerged from the analysis which could be implemented to increase the overall experience at the destination.

*4.5. A Proposed Fifth Domain of Tourism Experience*

Entrepreneur experience, in this study, refers to the entrepreneurs or business owners being the experience themselves, which occurs through consumers assimilating with them. Although not included in Pine and Gilmore's experience economy model, the role of the entrepreneur as the experience was a notable theme that emerged through the findings. This study proposes that the entrepreneur experience deserves its own standing as a separate experience domain, even though it can be noted that the entrepreneurs are also held responsible for staging the entertainment, educational, esthetic, and escapism experiences. The proposed framework is illustrated in Figure 2. Entrepreneur experience is supported by extant studies relating to tourists–host interactions which suggest that entrepreneurs and business owners are a crucial driver that enhances the tourist experience [84]. The creation of activities and the environment by the owner influences the individual experience. The

total experience depends on how such creations can capture the attention of and increase engagement from an individual [85].

In the case of the TBF, the craft beer entrepreneurs constitute one of the salient elements that attracted attendants to the festival. As festivals are events that bring brewers and craft beer brands together, it allows attendants to have close communication with these entrepreneurs [23]. Evidence from an online source showed that one Instagram account posted several images of herself with various brewers at TBF. This could imply that festival attendees were not only having a pleasant time from the entertainment and the surrounding atmosphere but were also engaged in a delightful experience connecting with the entrepreneurs. In addition, the interview data confirmed that meeting with craft beer entrepreneurs is the main motivator of the participants in attending the festival. The following excerpt presents the intention of one festival attendee:

> *I went to the festival to talk and exchange experiences with brewers to get updated. It's good that I can meet up with many brewers at the same event. I also want to support Thai craft beer brands.* (I14)

Because craft beer brewing is considered an artwork or a form of artisan entrepreneurship [31], many craft beer consumers become drawn to their favorite artists and have devoted themselves to brands. Attending a festival that has multiple brewers opens opportunities for the attendees to interact with and support their favorite craft brewers and brands. Hence, it was possible to witness many Thai craft beer brewers or owners standing by at their booths. Thus, the attendees could easily reach and interact with them, which led to the increasing of their satisfaction and revisiting intention.

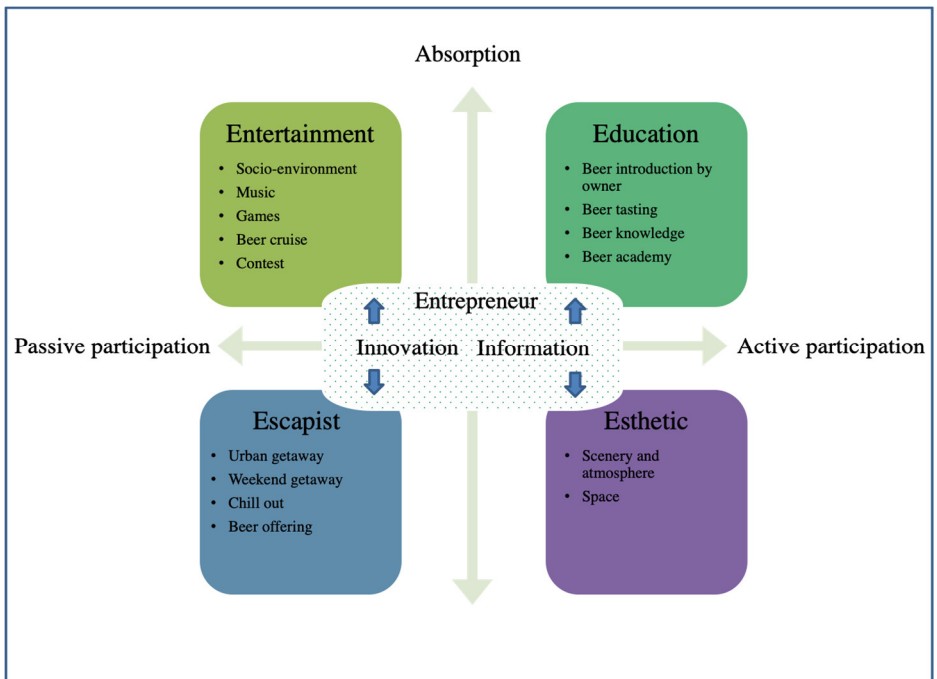

**Figure 2.** The proposed experience economy model with entrepreneur experience.

At CB, customers also perceived positive experiences from the owner, Chit, who is always present at the premises serving customers himself and recreating activities such as the games mentioned earlier. Many online reviewers highlighted the role of Chit as an important source of meaningful experience:

> *The best was that [Mr.] Chit was there behind the bar greeting and serving customers with the famous Thai smile.* (TA3)

> *It was nice to meet the owner of the establishment and get to know him.* (TA16)

> *They serve beer with pride and it tastes good. The staffs are informative and the owner Chit is very lively and homely.* (GR32)

In particular, Chit's friendly and approachable trait is a key driver in attracting general customers to CB. Based on the researcher's observation, Chit is always seen walking around the dining area greeting customers and sometimes sharing his stories at their tables. Key topics of discussions between Chit and customers often ranged from craft beer knowledge to Chit's role in revolutionizing the Thai craft beer movement and CB being a platform to connect and strengthen the Thai craft beer community.

On another level, CB also serves as a magnet for the more enthusiastic and specialized crowd of customers who have a desire to meet and engage with Chit at a personal level. A few online reviewers referred to how meeting Chit was like meeting the icon of Thai craft beer. As one reviewer stated, "You may run into 'Wichit' or 'Chit', the legendary man behind the local craft beer movement". Chit's own fame and willingness to support those interested in pursuing a hobby or business in craft beer brewing hence enhanced the overall experience of visiting CB. As Chit mentioned:

> *Chit Beer acts as a relationship broker for members of the craft beer community to meet. Our location helps filter out the visitors since you really have to make an effort to visit us. The end result is we have this extraordinary pool of individuals with different backgrounds who are eager to share ideas and learn from one another.* (I1)

The above findings illustrated the significant role of the entrepreneur as the experience. Entrepreneur experience was presented in both CB and the TBF. Entrepreneurs in the craft beer sector include the owners of craft beer brands and brewers which have an influence on productions and services. Entrepreneur experience at the two beer tourism forms occurred in similar manners. Firstly, tourists valued the experience from interacting with the entrepreneurs who were considered their icons or idols. Secondly, receiving a service directly from the owners increased the tourists' satisfaction which links to their overall experience. Lastly, acquiring information on the products from the entrepreneur, or horse's mouth, responded to one of the tourists' objectives on visiting the brewery and attending the festival. The cases confirmed how the role of entrepreneurs is essential, especially for small business, and can serve as a driving force that contributes to economic prosperity [86].

## 5. Conclusions and Recommendations

The popularity of craft beer has contributed to the development of craft beer tourism in many parts of the world as well as in Thailand. In Thailand, craft beer businesses have flourished in recent years, which is associated with the increase of craft beer tourism forms. However, the government laws and regulations have limited certain tourism activities which hinders service provision and growth in the business. As a result, small businesses are mostly affected by these constraints. To gain customer satisfaction and overcome business sustainability, craft beer entrepreneurs can take advantage of creating meaningful experiences as part of their business strategy. This study used a comparative analysis of two craft beer tourism forms to identify experience domains that can be applied to small businesses. The findings revealed that both craft beer tourism forms have practiced experience economy elements in their business. In the entertainment domain, several activities were offered to enhance the customer or tourist experience. Music, games, and socio-environmental space are among activities that increase the enjoyment of brewery customers and festival attendees. Educational experiences were provided through beer tasting, beer knowledge, and brewing lessons. The beer tasting and knowledge were presented in both brewery visits and beer festivals while the brewing lesson was a unique practice that can only be found at CB. Regarding the esthetic domain, the two craft beer tourism forms provide experience by utilizing dissimilar resources. The beer festival advanced man-made construction at the venue whereas CB benefited from the natural environment. In the case of the escapist dimension, both craft beer tourism forms used activities and atmosphere to offer elements that differ from everyday life. The TBF organized the festival atmosphere

which involved several activities and a wide variety of beers to provide extraordinary sentiment at the venue. Comparably, the relaxing atmosphere at CB allowed visitors to have weekend and urban getaways. Lastly, the entrepreneurial domain had an influence on both forms of craft beer tourism. Three entrepreneurial components, namely interacting with, acquiring information, and receiving service from the owner had a strong impact on consumers' or tourists' experience.

This study offers two main theoretical implications. First, it identified experiential domains in craft beer tourism using the experience economy framework. Even though this framework has been widely used to examine many types of tourism, it has not been studied in the craft beer tourism context. The evidence showed that craft beer tourism operators utilized several experience elements in their business which differ from other types of tourism. For instance, the variety of the product, which is the main characteristic of craft beer, can increase escapist experience value from the individual. Besides, such variety associate with product information and knowledge in which entrepreneurs can have an influence in providing the experience. Second, this study extended the traditional four Es in the experience economy. The findings revealed that an entrepreneur is also one of the significant domains that play a major role in the consumer or tourist experience. Craft beer entrepreneurs provide a magnetism that tourists desire to interact with. They are also the main factor that associates with tourist satisfaction and revisiting intention in craft beer tourism. The proposed experience economy model with the entrepreneur domain is exhibited in Figure 2.

From a managerial standpoint, this study exhibited the experience elements that were applied in two forms of craft beer tourism. In turn, other craft beer businesses, especially pubs and bars, can adopt these elements in their marketing strategies. To date, craft beer businesses are very competitive due to the increasing number of establishments. Therefore, in order to constantly grow and build a sustainable business, it is important to select the effective elements in the strategy to influence the visitors' satisfaction. The results of this study reveal different practical elements, which other craft beer businesses can duplicate some of to suit their operations. Given the niche and small characteristics of craft beer businesses, consumers or tourists can experience interacting with owners or brewers and acquire product information that links to multiple total experiences at the destination. Hence, other tourism businesses with similar features could use the entrepreneur experience domain.

As a practical recommendation, owners should devote more to the innovative and informative competencies and divert them to strategy to create a uniqueness of the tourism experience to constantly heighten the demands of the consumers, which leads to business sustainability. Hence this study proposes two craft beer entrepreneur competencies, namely innovativeness and informativeness, that can be developed to drive tourism experiences. Innovativeness is the ability to create new business opportunities by adopting creativity and novel techniques in products or services [87]. Prior research revealed that the owners of small businesses with an innovative mindset tend to be successful in providing consumer services [88]. In the craft beer sector, innovation and creativity are primary applications for business accomplishment. Many owners or brewers have put their creativity into the products to build brand characters and differentiation. However, good products alone cannot carry out total experience for the consumers. In the food tourism scheme, quality may not only refer to the taste and its consistency it also touches upon the experience value, which can exaggerate the tourists' satisfaction. Therefore, to generate the strategies, owners could inspire innovation and transcend traditional viewpoints when designing the experience dimensions in their businesses. Driven by the findings, innovation can be practiced in the entertainment and esthetic dimensions where individuals passively participate at the destination. For instance, craft beer establishments by the river could offer extra services such as craft beer cruises in which consumers and tourists would perceive new experiences during the cruise hours. Moreover, collaboration with other businesses could create unorthodox experience values. For example, craft brewers could collaborate

with a famous chef to brew a special batch of craft beer or design a beer-themed menu in which craft beers are paired with selected food according to their characteristics. The entertainment and esthetic experiences could be simultaneously initiated while tourists indulge in these recreative activities.

Informativeness as a competency is equally crucial to craft beer entrepreneurs in terms of constructing experience. Information competency is the ability to share information of the product details and production process by using an individual's knowledge [89]. In the food and beverage industry, consumers perceive the quality by searching for the information of such a product and experiencing the product through its taste and cooking attributes [52], thus, acquiring the information associated with the increasing of consumers' experience. Comparably, in the craft beer sector, consumers are likely to try new craft beers and are "curious to acquire information about [them]" [90] (p. 12). This extent aligns with the findings in the current study that craft beer tourists seek information while they travel to the brewery and participate in the festival. Therefore, information can be engaged in the education and escapist dimensions wherein the tourists actively search for new knowledge that cannot be acquired in common settings. Regarding the information strategy, owners or brewers should provide reliable and comprehensible information using an individual's knowledge and information and communication technologies (ICTs) in a parallel manner. When providing information on the products or the brewing process, it is essential that brewers explain the technical terms for a clear understanding. Accordingly, the information technologies, such as SMS, email, and social media platforms, should be established to inform about updates and future products that will be launched. Information technologies initiate two-way communication, which enables the engagement of the consumers. The information should be more extensive than the product knowledge, hence, owners could tell the background story of the brewery or craft beer business to communicate with consumers on the core business goals and brewing inspiration [91]. Plus, telling the rationale on how brewers come up with the creative name of their beers could amplify the education and escapist experiences of the tourists.

## 6. Limitations and Future Research

The findings in this study are subject to two limitations. First, the limitation lay in the data collection method. The lack of the online data used in the festival case is limited by the nature of social media posting. Many of the Facebook and Instagram posts display only pictures and missing textual comments. However, this study employs these platforms because the festival case does not appear on the same user-generated content platform as the brewpub case. Consequently, this links to the second limitation concerning the different sources of a sample, which may not be seamless for the comparative analysis study. In these regards, future research may consider other methods for collecting available data from the same source. The questionnaire may be used to collect the data in two cases because it could gain numbers of respondence in a short period, which is suited for the festival case. Besides, it could provide more complex relationships in the findings.

**Author Contributions:** Conceptualization, methodology, validation, formal analysis, investigation, data curation, writing—original draft preparation, R.C.; writing—review and editing, S.S.; visualization, R.C.; supervision, S.S. All authors have read and agreed to the published version of the manuscript.

**Funding:** This research was funded by Multicultural ASEAN Center Project (MU-MAC), Research Institute for languages and Cultures of Asia, Mahidol University, grant number 1/2018. The APC was funded by Mahidol University International College.

**Institutional Review Board Statement:** This research was approved by the Institute for Population and Social Research, Mahidol University Institutional Review Board, Protocol No. COA. 2019/06-204.

**Informed Consent Statement:** Informed consent was obtained from all subjects.

**Data Availability Statement:** Not applicable.

**Acknowledgments:** This research is part of Rangson Chirakranont's PhD thesis entitled "The Thai craft beer movement: tourism and consumption community", conducted under the affiliation of the Research Institute for Languages and Cultures of Asia, Mahidol University. The authors would like to thank Malinvisa Sakdiyakorn for her constructive feedback, and three anonymous reviewers for their valuable comments.

**Conflicts of Interest:** The authors declare no conflict of interest.

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
