# Peer review of "Applications of Experience Economy in Craft Beer Tourism: A Case Study in Thailand’s Context"

_sustainability, doi:10.3390/su131810448_

Round 1
Reviewer 1 Report
The manuscript entitled ‘Applications of experience economy in craft beer tourism: A case study in Thailand’s context’ presents interesting issue, however some corrections are needed
- Lines 8-9 – ‘Thailand as a context, this study explores the applications of Pine and Gilmore's experience economy framework in two forms of craft beer tourism, namely brewpubs and festivals’ – I am not native English language speaker, but the styling and grammatical quality of the writing in the manuscript impedes the understanding of the content in numerous situation.
- The structure of Abstract should be improved – the aim of the study, methodology (including number of respondents), main findings and implication should be presented.
- At the end of the Introduction section, authors should presented properly the aim of the study.
- Line 319 – ‘pictures were taken’ – please provide the legal regulation (approval from specific institution) for taking pictures of people.
- For the research that involves human subjects the rules of the Declaration of Helsinki of 1975 must be applied, including ethics commission approval and especially informed consent. Please add the information about number of ethics commission approval (specific reference)
- How was informed consent of the respondents collected?
- Line 323 - total of 25 interviews and informal conversations’ – how these respondents were selected? Please provide the scenario of interviews (in appendix)
- Lines 339-340 –‘203 reviews were collected which derived from 28 reviews on TripAdvisor and 175 reviews on Google review’ – please provide the criteria of selection of these reviews
- Line 349-350 – ‘Online reviews were in Thai and English, and was transcribed immediately as well.’ Who translated the review? Please provide the information.
My major concern is associated with data and data collection. Authors did not present the proper scientific approach when gathering the data, so the results are biased. The interviews methodology are not describe properly and involving small number of respondents (sample size), whereas reviews from TripAdvisor or Google reviews are not good and reliable source of information.
Reviewer 2 Report
In the manuscript the authors presented an important point, although the manuscript has some drawbacks.
Main remarks:
1. Intruduction - what is the research gap?
2. Materials and Methods - in my opinion, there are no in-depth taxonomic and statistical methods. In my opinion, it is worth using such methods, especially since own empirical research was conducted.
3. Conclusions - in the conclusions, I propose to write what is new to the manuscript and whether the purpose of the manuscript has been achieved.
4. Other comments:
- Incorrect manuscript footnotes, see editorial requirements: e.g. lines 447, 450, 520 ...
- in my opinion, there is no requirement to include page numbers in the text: e.g. lines 317, 523 ...
Reviewer 3 Report
The paper " Applications of experience economy in craft beer tourism: A case study in Thailand’s context" is very interesting and useful to understand some dynamics related to craft beer tourism in four experience economy dimensions i.e. Entertainment, Education, Escapist, Esthetic. The paper is clear, the contents are well explained and easy to read. Findings obtained by the authors also highlight the important role of the entrepreneur in craft beer tourism experience.
The introduction is clear and provides information to understand the objectives of the study and the literature review is logical with the treated concerns. In any case, I believe that the two sections need integrations to improve the state of art of the international craft beer phenomenon (introduction) and emphasize the linkage of the paper with the aims of the Sustainability journal (introduction and literature review) that, in my humble opinion, currently is not so evident.
Therefore, to improve introduction and literature review, please see for example the following papers:
Bahl, H.C., Gupta, J.N.D., Elzinga, K.G. A framework for a sustainable craft beer supply chain. (2021) International Journal of Wine Business Research, 33 (3), pp. 394-410. DOI: 10.1108/IJWBR-08-2020-0038
Bråtå, H.O. Local traditions as a means for commercial production of historical beers: The case of Vossaøl, Norway. (2017) Norsk Geografisk Tidsskrift, 71 (5), pp. 301-312. DOI: 10.1080/00291951.2017.1395909
Cabras, I., Ellison, K. Craft beers and beer festivals: Exploring the potential for local economies and gastro-tourism in the UK. (2018) Gastronomy and Local Development: The Quality of Products, Places and Experiences, pp. 193-213. DOI: 10.4324/9781315188713
Cortese, D., Pairotti, M.B., Giachino, C., Bonadonna, A. The art of craft beer in a traditional wine nation: A preliminary study in a model area. (2017) Quality - Access to Success, 18 (161), pp. 149-156.
de Jong, A., Steadman, C. (Re)crafting belonging: cultural-led regeneration, territorialization and craft beer events. (2021) Social and Cultural Geography, DOI: 10.1080/14649365.2021.1939126
De Simone, N., Russo, P., Tufariello, M., Fragasso, M., Solimando, M., Capozzi, V., Grieco, F., Spano, G. Autochthonous biological resources for the production of regional craft beers: Exploring possible contributions of cereals, hops, microbes, and other ingredients. (2021) Foods, 10 (8), DOI: 10.3390/foods10081831
Duarte Alonso, A., Alexander, N. Craft Beer Tourism Development “Down Under”: Perspectives of Two Stakeholder Groups. (2017) Tourism Planning and Development, 14 (4), pp. 567-584. DOI: 10.1080/21568316.2017.1303541
Duarte Alonso, A., Kok, S., O'Shea, M. Peru's emerging craft-brewing industry and its implications for tourism. (2021) International Journal of Tourism Research, 23 (3), pp. 319-331. DOI: 10.1002/jtr.2408
Dunn, A., Kregor, G.Craft beer festivals: The craft brewers' experience of getting beer to market in the USA and UK. (2017) Exhibitions, Trade Fairs and Industrial Events, pp. 145-159. DOI: 10.4324/9781315415291
Embry, E.A. Green beer: Why small to medium sized enterprises adopt sustainable practices. (2018) 78th Annual Meeting of the Academy of Management, AOM 2018, . DOI: 10.5465/AMBPP.2018.105
Gajić, T., Raljić, J.P., Blešić, I., Aleksić, M., Vukolić, D., Petrović, M.D., Yakovenko, N.V., Sikimić, V. Creating opportunities for the development of craft beer tourism in serbia as a new form of sustainable tourism. (2021) Sustainability (Switzerland), 13 (16), DOI: 10.3390/su13168730
Gatrell, J., Reid, N., Steiger, T.L. Branding spaces: Place, region, sustainability and the American craft beer industry. (2018) Applied Geography, 90, pp. 360-370. DOI: 10.1016/j.apgeog.2017.02.012
Hodge, M.G., Torsney, B.M., Paris, J.H. Ticket to intoxication: Exploring attendees’ motivations for attending craft beer events. (2021) Leisure Studies, DOI: 10.1080/02614367.2021.1948591
Mac an Bhaird, C., Owen, R., Dodd, S.D., Wilson, J., Bisignano, A. Small beer? peer-to-peer lending in the craft beer sector. (2019) Strategic Change, 28 (1), pp. 59-68. DOI: 10.1002/jsc.2246
Ness, B. Beyond the Pale (Ale): An exploration of the sustainability priorities and innovative measures in the craft beer sector. (2018) Sustainability (Switzerland), 10 (11), art. no. 4108, DOI: 10.3390/su10114108
Nesse, K., Green, T., Ferguson, B. Quality of life in potential expansion locations is important to craft brewers (2019) Journal of Regional Analysis and Policy, 49 (1), pp. 65-77.
Niemi, L., Kantola, J. Legitimated consumption: a socially embedded challenge for entrepreneurs’ value creation. (2018) Journal of Research in Marketing and Entrepreneurship, 20 (2), pp. 214-228. DOI: 10.1108/JRME-10-2016-0038
Pachura, P. The role of space in the business models of microbreweries. (2020) Applied Geography, 125 DOI: 10.1016/j.apgeog.2020.102303
Reid, N. Craft Beer Tourism: The Search for Authenticity, Diversity, and Great Beer (2021) Advances in Spatial Science, pp. 317-337. DOI: 10.1007/978-3-030-61274-0_16
Rogerson, C.M., Collins, K.J.E. Entrepreneurs in craft beer and tourism: Perspectives from South Africa. (2019) Geojournal of Tourism and Geosites, 27 (4), pp. 1158-1172. DOI: 10.30892/gtg.27404-423
Schroeder, S. Crafting New Lifestyles and Urban Places: The Craft Beer Scene of Berlin. (2020) Papers in Applied Geography, 6 (3), pp. 204-221. DOI: 10.1080/23754931.2020.1776149
Sforzi, J., Colombo, L.A. New opportunities for work integration in rural areas: The 'social flavour' of craft beer in Italy. (2020) Sustainability (Switzerland), 12 (16), art. no. 6351, . DOI: 10.3390/SU12166351
Sjölander-Lindqvist, A., Skoglund, W., Laven, D. Craft beer – building social terroir through connecting people, place and business. (2020) Journal of Place Management and Development, 13 (2), pp. 149-162. DOI: 10.1108/JPMD-01-2019-0001
Skoglund, W., Selander, J. The Swedish alcohol monopoly: A bottleneck for microbrewers in Sweden? (2021) Cogent Social Sciences, 7 (1), art. no. 1953769, DOI: 10.1080/23311886.2021.1953769
Strohacker, K., Fitzhugh, E.C., Wozencroft, A., Ferrara, P.-M.M., Beaumont, C.T. Promotion of leisure-time physical activity by craft breweries in Knoxville, Tennessee. (2021) Leisure Studies, DOI: 10.1080/02614367.2021.1933574
Wojtyra, B., Kossowski, T.M., BĹ™ezinová, M., Savov, R., LanÄŤariÄŤ, D. Geography of craft breweries in Central Europe: Location factors and the spatial dependence effect. (2020) Applied Geography, 124, DOI: 10.1016/j.apgeog.2020.102325
Finally, Research methods, Findings and discussion, Conclusions and recommendations are clear, well explained and provide information that permit to easily understand what has been done and achieved. Discussion and Conclusions could be reformulated in line with the modifications that will be made. Overall, I appreciate the topic of your paper and I consider your study interesting and quite original.
Round 2
Reviewer 1 Report
The authors corrected or explained most of my comments. However, I think that there is still room for improvement.
Reviewer 2 Report
Accept in present form